# Ag Nanocluster Production through DC Magnetron Sputtering and Inert Gas Condensation: A Study of Structural, Kelvin Probe Force Microscopy, and Optical Properties

**DOI:** 10.3390/nano13202758

**Published:** 2023-10-13

**Authors:** Ishaq Musa, Naser Qamhieh, Saleh T. Mahmoud

**Affiliations:** 1Department of Physics, Palestine Technical University-Kadoorie, Tulkarem P.O. Box 7, Palestine; 2Department of Physics, UAE University, Al-Ain P.O. Box 15551, United Arab Emirates; nqamhieh@uaeu.ac.ae

**Keywords:** silver nanoclusters, DC magnetron sputtering, Kelvin Probe Force Microscopy (KPFM), UV-visible absorption, photoluminescence

## Abstract

Silver nanoclusters are valuable for a variety of applications. A combination of direct current (DC) magnetron sputtering and inert gas condensation methods, employed within an ultra-high vacuum (UHV) system, was used to generate Ag nanoclusters with an average size of 4 nm. Various analytical techniques, including Scanning Probe Microscopy (SPM), X-ray Diffraction (XRD), Kelvin Probe Force Microscopy (KPFM), UV-visible absorption, and Photoluminescence, were employed to characterize the produced Ag nanoclusters. AFM topographic imaging revealed spherical nanoparticles with sizes ranging from 3 to 6 nm, corroborating data from a quadrupole mass filter (QMF). The XRD analysis verified the simple cubic structure of the Ag nanoclusters. The surface potential was assessed using KPFM, from which the work function was calculated with a reference highly ordered pyrolytic graphite (HOPG). The UV-visible absorption spectra displayed peaks within the 350–750 nm wavelength range, with a strong absorption feature at 475 nm. Additionally, lower excitation wavelengths resulted in a sharp peak emission at 370 nm, which became weaker and broader when higher excitation wavelengths were used.

## 1. Introduction

Nanoparticles, particularly silver (Ag) nano clusters, have garnered significant attention due to their exceptional properties, including a high conductivity, catalytic activity, and antimicrobial behavior [1,2]. These unique characteristics make Ag nano clusters promising candidates for various applications in fields such as electronics, optics, catalysis, and biomedicine. Silver nanoparticles (AgNPs) have unique and tunable optical properties. These properties arise primarily from the interaction of light with the free electrons present in the silver, leading to collective oscillations known as surface plasmon resonances (SPRs) [3]. These SPRs are responsible for the strong absorption and scattering of light by AgNPs, making them of particular interest in various applications, ranging from photonics and sensing to medicine and catalysis [4,5]. The optical behavior of AgNPs is highly dependent on their size, shape, and surrounding environment [6]. For instance, small spherical nanoparticles primarily exhibit a single SPR band in the visible region, but as their size or shape changes, the resonance can shift to different wavelengths or even split into multiple bands [7]. This tunability allows for the design of nanoparticles with specific optical characteristics tailored for particular applications. Furthermore, the proximity and interaction of AgNPs with other materials, such as dielectrics or other metal nanoparticles, can lead to coupled plasmonic modes, further enriching the optical landscape. These interactions have been harnessed in the development of advanced materials and devices, such as plasmonic sensors that can detect minute changes in the local environment or hybrid structures that enhance light–matter interactions.

However, synthesizing Ag nano clusters with precise control over their size, shape, and composition remains a key challenge. Several methods have been developed, such as chemical reduction [8], laser ablation [9], and thermal evaporation [10], but these techniques often suffer from limitations in terms of complex procedures, harsh reaction conditions, or inadequate control over nanoparticle characteristics [11]. DC magnetron sputtering and inert gas condensation has emerged as a versatile and promising technique for Ag nano cluster synthesis. It involves a physical vapor deposition process where a silver target is bombarded with high-energy ions, causing atoms or clusters to be released and subsequently condense onto a substrate, forming the desired nanoparticles. By carefully adjusting process parameters such as pressure, temperature, and deposition time, precise control over the properties of the resulting Ag nano clusters can be achieved. The advantages of using DC magnetron sputtering for Ag nano cluster synthesis include its ability to produce nanoparticles with a controlled size, shape, and density [12,13,14,15]. The inert gas condensation (IGC) technique is a promising approach for producing high-quality nanoparticles with controlled characteristics. This technique involves the evaporation of a source material into an inert gas, followed by its controlled condensation, leading to the formation of nanoparticles. While the fundamental principles of IGC are well-understood, optimizing the process parameters for specific materials and desired nanoparticle characteristics remains a challenge [16]. Gas-phase synthesis technology has rapidly evolved in recent years, allowing for the fabrication of complex nanoparticles with a controllable chemical composition and structure.

Gas-phase synthesis technology offers several advantages for the fabrication of complex nanoparticles with a controllable chemical composition and structure. It allows for size control and flexibility in material choice, making it possible to synthesize multi-element nanoparticles with independent control over core size and shell thickness [17]. Gas-phase methods also provide a more flexible choice of materials and avoid the use of solvents, which is advantageous for applications in catalysis and optoelectronics [18]. Additionally, gas-phase synthesis methods can generate nanoparticles with both a uniform size and composition, as well as a uniform bimetallic configuration, by incorporating heat treatment steps [19]. The use of an inert gas ensures that the synthesized nanoparticles are free from contaminants. The manipulation of the process parameters, such as the choice of target material, gas composition, pressure, and power, enables precise control over the nucleation and growth of Ag nano clusters. Moreover, this technique allows for deposition onto various substrates, such as silicon, glass, mica sheet, or flexible polymers, facilitating integration into different devices and applications [20]. DC magnetron sputtering is an effective method for depositing thin films and nanostructures, offering control over the deposition parameters to achieve precise control over the cluster size. The focus of this study is to delve into the synthesis of silver (Ag) nanoclusters using the DC magnetron sputtering technique. The primary objective is to comprehensively examine the morphology, structural characteristics, Kelvin probe force microscopy (KPFM) measurements, and optical properties of the resulting nanoclusters. The morphology of the synthesized Ag nanoclusters will be studied using scanning probe microscopy (SPM), while their structural properties will be investigated through an X-ray diffraction (XRD) analysis. The optical behavior of the nanoclusters will be analyzed using UV-Visible spectroscopy and photoluminescence. Additionally, KPFM will be employed to evaluate the electrical properties of the nanoclusters. Within the context of this study, the utilization of KPFM will provide valuable insights into the electrical properties of the synthesized silver nanoclusters. This characterization technique will yield information about their surface potential, work function, charge distribution, and any variations in their electrical behavior. The investigation of these properties will enhance our understanding of the electronic properties of the nanoclusters and their potential applications in electronic devices, energy storage, and other relevant fields. The comprehensive nature of this research aims to contribute to the development of novel applications in areas such as catalysis, sensing, and optoelectronics by providing valuable insights into the synthesis and characterization of Ag nanoclusters.

## 2. Materials and Methods

The deposition of silver nanoclusters onto various substrates, including Si, mica sheets, and glass, was accomplished inside a highly compatible ultra-high vacuum system (Nanogen-50, Mantis Deposition Ltd., Oxfordshire, UK). This involved magnetron sputtering and plasma-gas condensation techniques. In Figure 1, we present a schematic representation of our laboratory’s experimental setup for producing Ag nanoparticles. This diagram highlights key components such as a DC magnetron sputtering device, which serves as the nanoparticle generator, along with a turbo pump (TP), a quadrupole mass filter (QMF), and the chamber where deposition occurs. To achieve the necessary base pressure, we employed two turbo pumps for both the main and source chambers. The DC magnetron mechanism facilitated the production of nanoparticles from the Ag target. We introduced Argon gas to initiate the plasma, facilitate the material sputtering from its designated target, support the nanoparticle condensation, and establish a pressure difference between the source and deposition chambers, enabling the nanoparticles to traverse the mass filter. Notably, the mass filter in our study is a quadrupole type, strategically positioned between the source of the nanoparticles and the deposition chamber [21,22]. The entire procedure was conducted within an ultra-high vacuum (UHV) compatible system from (Mantis Deposition Ltd., Oxfordshire, UK), carefully maintained at an extremely low pressure of approximately 10-8 mbar, employing turbo pumps and a dry rotary pump. This meticulous control of the ultra-low-pressure environment played a critical role in ensuring the deposition process’s precision and accuracy. The deposition process relied on magnetron sputtering and inert gas condensation techniques. A high-purity silver target with a 99.99% purity was securely affixed to the magnetron sputter head. By applying a direct current (DC) discharge power and utilizing argon (Ar) gas as an inert gas, plasma was generated within the source chamber. This plasma facilitated the sputtering of silver atoms from the target surface.

The distance between the silver target and the exit nozzles of the source chamber, known as the aggregation length (L), was fixed at 40 mm. This parameter played a significant role in controlling the size characteristics of the resulting silver nanoclusters. By adjusting the aggregation length, we could manipulate the growth and aggregation of the nanoclusters, thereby influencing their size and distribution [23]. The choice of fabrication parameters for the Ag nanoclusters, including a discharge power density of 50 W/cm^2^, a U/V ratio of 0.09, and an aggregation length of 40 mm, was based on previous work [23,24].

To induce the condensation of the silver material and promote nanocluster formation, a controlled flow of argon gas was maintained at a rate of 60 standard cubic centimeters per minute (sccm), regulated using an MKS mass flow controller from (Andover, MA, USA). The argon gas served a dual purpose: assisting in the sputtering process and acting as a carrier gas for the condensation of the silver atoms into nanoclusters.

The DC discharge power supplied during the process was measured at 50 W. This power level played a crucial role in determining the size and characteristics of the resulting nanoclusters due to modulating this power level to exert control over the growth and nucleation processes [25].

Before depositing the silver nanoclusters onto the target surface, we utilized a quartz crystal monitor (QCM) to precisely determine the rate of deposition. The QCM was mounted on a linear translator and strategically positioned to approach the nanocluster beam, allowing us to obtain accurate measurements before retracting it to its initial position. Furthermore, we employed a quadrupole mass filter (QMF) to analyze the size distribution of the nanoclusters. The QMF operates on the principle of using both alternating current (AC) and direct current (DC) voltages, denoted as [±(U + Vcos ωt)], across four linear metal rods. Of these, two rods are linked to a positive voltage and the remaining two are connected to a negative voltage. In this context, U represents the DC voltage, V signifies the AC voltage’s amplitude, ω stands for frequency, and t denotes time.

Additionally, an external grid is attached to the mass filter to gauge the ion flow of a specific mass/size. The subsequent current is then recorded using a pico-ampere meter [21].

The size and morphology of the Ag nanoclusters were assessed using Scanning Prob microscopy (SPM-9700HT, Shimadzu, Tokyo, Japan). The topographical representations were meticulously captured using a specialized mode known as the non-contact dynamic mode in atomic force microscopy (AFM). This mode is particularly beneficial as it minimizes the physical contact between the AFM tip and the sample, thereby reducing potential damage or alteration to the sample’s surface. For these precise measurements, we employed an AFM tip from a renowned manufacturer, Budget Sensor. The specific model of the tip used was the SPP-NCHR. This model is notable for its distinct specifications that make it fitting for high-resolution imaging. It possesses a force constant, which is essentially a measure of the tip’s stiffness, of 42 N/m. This force constant ensures that the tip responds appropriately to the surface’s contours without applying excessive force. Furthermore, the tip has a resonance frequency of 330 kHz. The resonance frequency is a critical parameter in AFM, as it determines the natural frequency at which the cantilever vibrates. A higher resonance frequency, like the one in the SPP-NCHR model, often translates to faster imaging speeds and a better sensitivity. Lastly, one of the most crucial aspects of this tip is its sharpness. The SPP-NCHR model boasts a tip with an ultra-fine radius of less than 10 nm. This sharp tip ensures that the images captured are of the highest resolution. To study the surface potential and work function of the Ag nanocluster, we employed a combination of atomic force microscopy (AFM) and Kelvin probe force microscopy (KPFM). These techniques allowed us to precisely measure both the surface topography and the surface potential of the Ag nanocluster. The sample was securely positioned on a piezo-stage, ready for scanning. A conductive AFM probe made of PtSi was affixed to the cantilever’s tip. As this probe approached and moved across the sample surface, it maintained a consistent oscillating force, producing a detailed topographic image of the surface. KPFM further enhanced our investigation by allowing a simultaneous acquisition of the surface potential image and topographic data at every point of measurement. When the probe is in close proximity to the sample surface, the force exerted on it includes an electrostatic component. This force arises due to the contact potential difference (CPD) between the probe and the sample, providing insights into the surface potential [26]. For accurate imaging of the topography and potential at the Ag nanocluster, careful preparation of the experimental samples was essential. The sample was securely adhered to a carbon tape, which was then firmly attached to a steel disk. This ensured stability during the scanning process. The measurements were conducted with a scanning speed set at 0.3 Hz and a high resolution of 256 × 256 pixels, ensuring detailed and clear images. The spatial resolution of the device was determined to be 0.2 nm. The X-ray diffraction analysis was conducted using a Shimadzu (Kyoto, Japan) 6100 XRD instrument equipped with CuKα radiation (λ = 0.15406 nm). The UV-visible absorption spectra of the samples were measured on a UV-2600i (Shimadzu, Tokyo, Japan) spectrophotometer. The photoluminescence spectra were measured using an RF-6000 spectrofluorometer (Shimadzu, Tokyo, Japan). The Shimadzu Lab solution software was used for data acquisition.

## 3. Results and Discussion

### 3.1. Morphology and Structure

Figure 2A displays the AFM topographic image of the Ag nanoclusters deposited on a mica substrate, clearly indicating their spherical shape. A three-dimensional projection of the nanocluster is depicted in Figure 2B. A height analysis, as shown in Figure 2C, reveals that the observed heights of these nanoclusters approximately range from 3.5 nm to 4.5 nm. Figure 2D presents a particle size distribution histogram for the Ag nanoclusters, based on data extracted from approximately 1400 nanoparticles in the image in Figure 2A. An analysis using particle measurement software indicates that the average diameter of these nanoparticles is around 4 nm.

Additionally, the solid line in Figure 3 represents a Gaussian distribution that fits the sizes of the Ag nanoclusters, as measured by the quadrupole mass filter in the nanocluster production system. This fit was obtained under specific conditions: a discharge power of 50 W, an Argon flow rate of 60 sccm, a chamber pressure of 1.17 × 10^−3^ mbar, a U/V ratio of 0.09, and an aggregation length of 40 mm. The graph indicates a peak diameter of approximately 4 nm, corroborating the size distribution previously identified through AFM topography in Figure 2D.

Figure 4 displays the X-ray diffraction (XRD) patterns for the Ag nanoclusters, verifying their face-centered cubic lattice structure. All the Ag nanoclusters display consistent diffraction profiles, featuring distinct peaks at 2θ angles of 37.7°, 43.9°, 64.2°, 77.4°, and 79.87°. These peaks correspond to the (111), (200), (220), (311), and (222) crystallographic planes of face-centered cubic crystals, respectively [27].

### 3.2. Kelvin Probe Force Microscopy

To explore the Kelvin Probe Force Microscopy (KPFM) of the silver nanoclusters on a silicon substrate, we used a conductive Atomic Force Microscopy (AFM) probe made of platinum silicide (PtSi). The probe comes into contact with the sample and scans its surface using a constant force oscillation. This process yields a topographic image that captures both the height and electrical potential of the surface. The contact potential difference (CPD) between the probe and the sample is derived from the potential measurement image. Figure 5 displays the AFM topography and KPFM images for a silver nanocluster. In Figure 5B, a high potential for the nanoparticles is represented in red, while A low potential is indicated in A yellow color. According to the line analysis of THE potential heights in Figure 5D, the average CPD was about 42 mV. Figure 6 displays the Atomic Force Microscopy (AFM) topographical features and surface potential of a reference sample of Highly Ordered Pyrolytic Graphite (HOPG), grade ZYA. This sample has a mosaic spread of 0.4 ± 0.1° and dimensions of 10 mm × 10 mm with a 1 mm thickness, supplied by MikroMasch. The height profile for the HOPG sample is approximately 0.15 nm, while the average Contact Potential Difference (CPD) is around 80 mV, as illustrated in Figure 6C and Figure 6D, respectively.

The goal of Kelvin Probe Force Microscopy (KPFM) is to accurately quantify the work function of a given sample. In the context of KPFM, the surface potential is a critical parameter, which is essentially the difference in the work function between the sample surface, denoted as $\emptyset_{sample}$, and the AFM probe tip, denoted as $\emptyset_{tip}$. In mathematical terms, this interaction between the probe tip and the sample surface can be expressed by equation:(1)$$\emptyset_{sample}=\emptyset_{tip}-eV_{CPD}$$
where (*e*) is the elementary charge and $V_{CPD}$ is the Contact Potential Difference between the sample and the tip [28]. To determine the work function of the Ag nanoparticles, it is essential to measure the work function of the cantilever used in the system. Calibration is crucial for accurate results and is typically performed using a reference sample with a well-known work function. In our case, we employ a Highly Oriented Pyrolytic Graphite (HOPG) sample for this purpose, as shown in Figure 6. The work function for (HOPG) typically lies between 4.5 and 5 eV [29,30,31]. Using Equation (1), we can formulate analogous equations for both Ag nanoparticles and HOPG substrates. If we represent the work function of the silver nanocluster substrate as $\emptyset_{Ag}\;$ and the contact potential difference specific to the silver nanocluster substrate as $V_{\left(CPD,Ag\right)},\;$ then the equation for the silver nanocluster substrate becomes:(2)$${\;\emptyset }_{Ag}=\emptyset_{tip}-eV_{\left(CPD,Ag\right)}$$

Similarly, denoting the work function of the HOPG substrate as ${\;\emptyset }_{HOPG}$ and its specific contact potential difference as $V_{\left(CPD,HOPG\right)}$, the equation for the HOPG substrate is:(3)$${\;\emptyset }_{HOPG}=\emptyset_{tip}-eV_{\left(CPD,HOPG\right)}$$

To find the difference between the two equations, we can subtract Equation (3) from Equation (2):(4)$${\;\emptyset }_{Ag}=\emptyset_{HOPG}+e(V_{\left(CPD,HOPG\right)}-V_{\left(CPD,Ag\right)})$$

Therefore, Equation (4) establishes a relationship between the work functions of the silver and HOPG by connecting them through their respective Contact Potential Differences. Using Equation (4) and the Contact Potential Difference (CPD) data sourced from Figure 5D and Figure 6D, referencing the work function value of HOPG from the literature [32], we determined the work function of the AgNPs that have an average diameter of 4 nm:(5)$${\;\emptyset }_{Ag}=4.55eV+e(0.08V-0.043V)=4.587V$$

From the literature, the work function of bulk silver is 4.3 eV [33], Ag(111) is 4.64 eV, and a vacuum-deposited Ag electrode is 4.68 eV [26,34]. The work function of the Ag nanoclusters is estimated to be around 4.59 eV, aligning closely with the values reported in the literature. Generally, a reduction in nanoparticle size can result in quantum confinement effects, which, in turn, can alter electronic characteristics, including the work function. Consequently, the elevated 4.59 eV value may serve as an indicator of such size-dependent variations.

### 3.3. Optical Properties

Silver nanoclusters possess distinct optical characteristics that set them apart in the realm of nanomaterials, making them a focal point in interdisciplinary research. Unlike bulk silver, these nanoclusters display a localized surface plasmon resonance (LSPR) effect, where incident light induces a collective oscillation of free electrons. This LSPR is highly sensitive to the nanocluster’s dimensions and geometry, allowing for precise control over its optical behavior [7,35]. Figure 7 shows the UV-visible spectrum of a Ag nanocluster. In this figure, a strong broad peak is observed at 470 nm. Ag nanoclusters are known to reveal a UV-visible absorption maximum in the range of 350–750 nm due to surface plasmon resonance [36]. This broadening of peak resonance in the absorption spectrum is due to increased damping effects, which can arise from various factors such as electron scattering, electron–phonon interactions, and increased heterogeneity in larger particles [37].

The photoluminescent characteristics of the silver nanoclusters were examined to identify the source of their fluorescence. It is noteworthy that these silver nanoclusters demonstrated fluorescence that changed based on the wavelength of the excitation light. As illustrated in Figure 8, the spectra displayed a distinct sharp peak at 370 nm, and broader peaks at 475 nm, 650 nm, and 737 nm. The peaks at 370 nm and 475 nm aligned with the surface plasmon resonance seen through the UV-visible spectroscopy, suggesting that the fluorescence primarily originates from single-electron transitions between specific energy states [38,39]. To further investigate, fluorescence emission data were collected at five selected excitation wavelengths: 210 nm, 220 nm, 230 nm, 250 nm, and 270 nm. The analysis revealed the presence of several fluorescence emission peaks. When the Ag nanocluster sample is excited with shorter wavelengths (210 nm, 220 nm, and 230 nm), a prominent and sharp emission peak emerges at 370 nm, accompanied by weaker, broader peaks at 475 nm, 650 nm, and 737 nm. In contrast, excitation with longer wavelengths (250 nm and 270 nm) weakens the resonance peak at 370 nm, while intensifying and broadening the other resonance peaks [38]. This variation in excitation and emission is attributed to the presence of both small and large nanoparticles, as well as aggregating nanoclusters. Typically, localized surface plasmon resonance (LSPR) frequencies are blue-shifted in smaller particles. As the nanoparticle size increases, the LSPR frequency shifts to lower frequencies or longer wavelengths. This shift is often due to the larger size facilitating more collective, less confined electron oscillations, resulting in a red-shift of the resonance peaks [40].

## 4. Conclusions

We successfully generated silver nanoclusters with an average diameter of 4 nm by employing a hybrid approach that combined DC magnetron sputtering and inert gas condensation techniques. An X-ray Diffraction (XRD) analysis confirmed the nanoclusters’ simple cubic structure, adding to our understanding of their geometric properties. Utilizing Kelvin Probe Force Microscopy (KPFM), we assessed the surface potential of these nanoclusters and calculated their work function, using a Highly Oriented Pyrolytic Graphite (HOPG) sample as a calibration reference. This information on work function is crucial for the effective integration of these nanoclusters into electronic devices such as transistors and photovoltaic cells. The UV-visible absorption studies indicated absorption peaks in the 350–750 nm wavelength range, with a particularly strong absorption feature at 475 nm. Furthermore, we observed wavelength-dependent emission characteristics; specifically, a sharp emission peak at 370 nm was prominent when excited at lower wavelengths, but became weaker and broader at higher excitation wavelengths. The optical behavior of silver nanoclusters makes them a promising material for various applications in nanotechnology and photonics.

## Figures and Tables

**Figure 1 nanomaterials-13-02758-f001:**
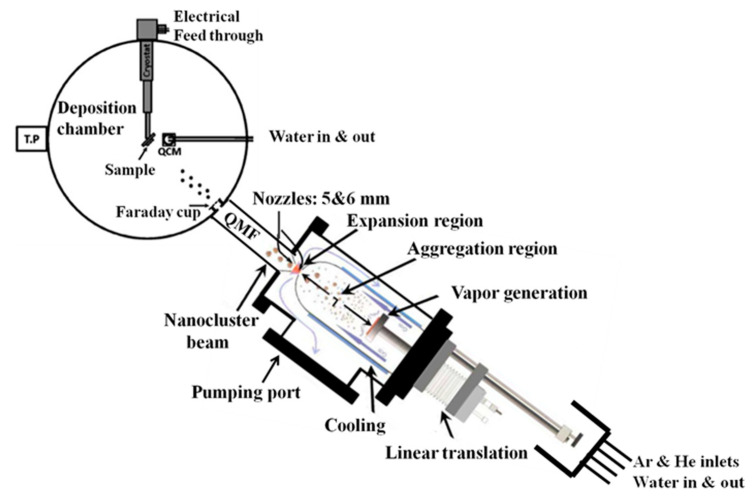
Diagram illustrating the deposition system, encompassing the nanoparticle source and the deposition chamber [22].

**Figure 2 nanomaterials-13-02758-f002:**
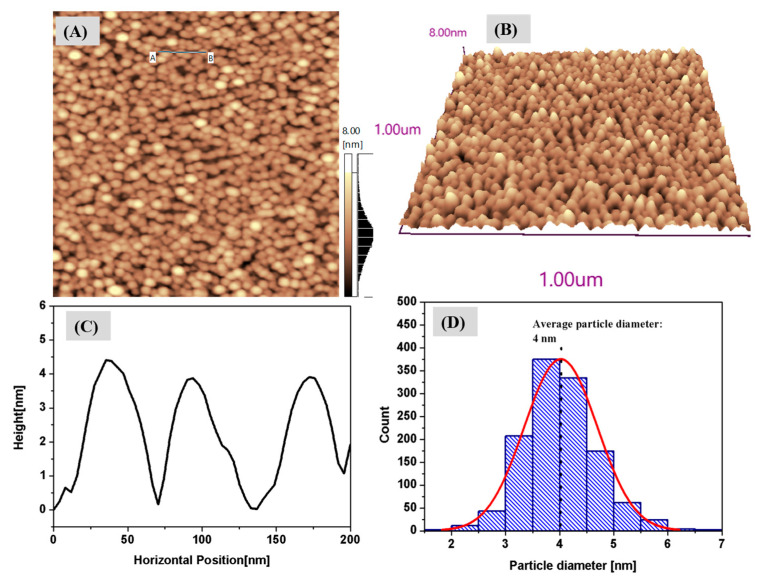
(**A**) Atomic force microscopy (AFM) topography images of Ag nanoclusters on a mica substrate, Scan sizes of 1 µm × 1 µm (**A**). (**B**) Three-dimensional projection of nanoclusters, (**C**) line profiles for nanoclusters identified in the image (**A**). (**D**) Particle size distribution histogram.

**Figure 3 nanomaterials-13-02758-f003:**
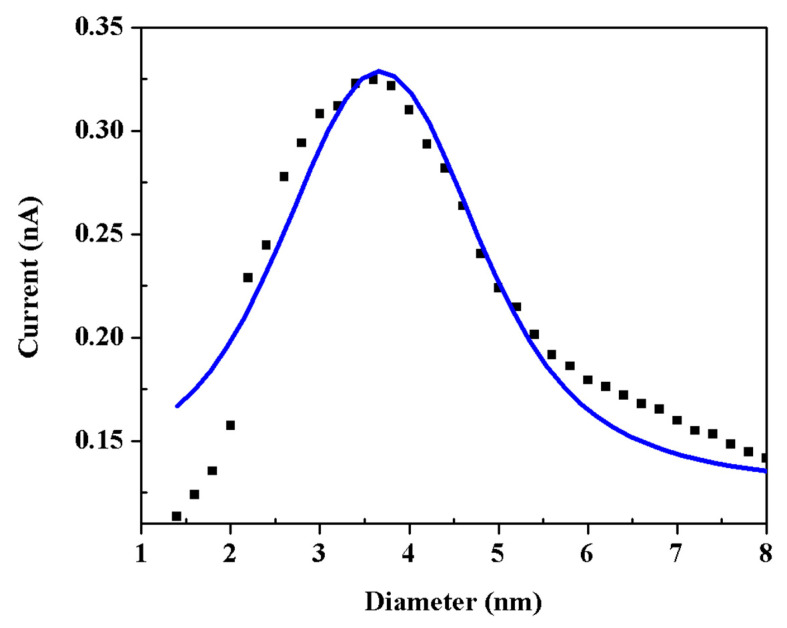
The Gaussian fit data solid line is the size distribution of Ag nanocluster using a quadrupole mass filter.

**Figure 4 nanomaterials-13-02758-f004:**
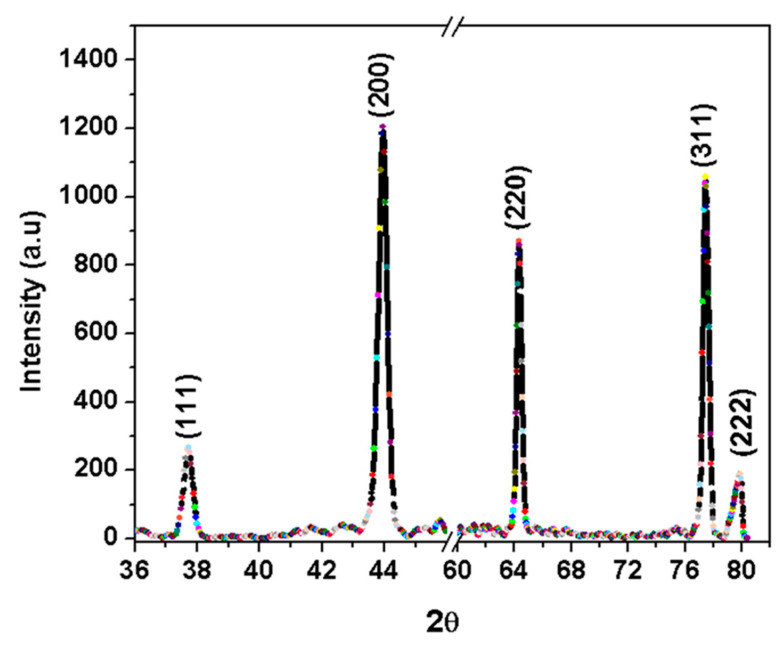
X-ray diffraction pattern of the silver nanocluster.

**Figure 5 nanomaterials-13-02758-f005:**
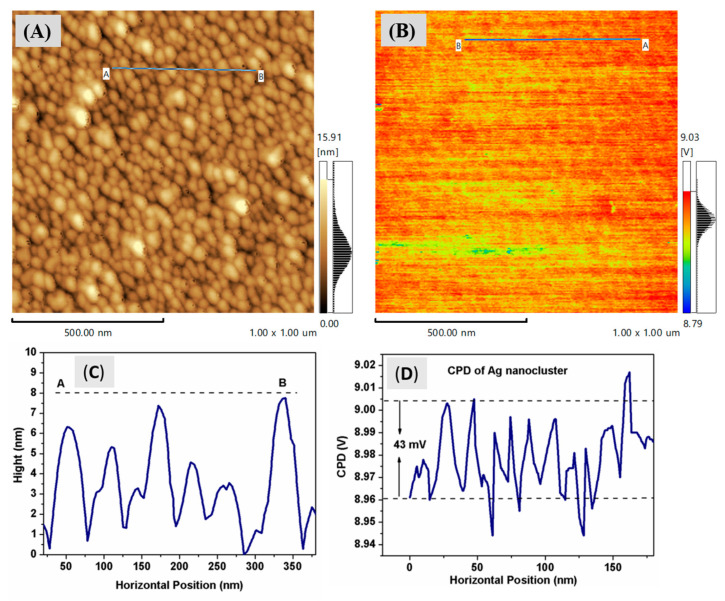
(**A**,**B**) Ag Nanocluster and surface potential simultaneously obtained topography images on the Si substrate, (**C**) line profiles for nanoclusters identified in the image (**A**,**D**) CPD profile measurements extract from image (**B**) between two points A and B.

**Figure 6 nanomaterials-13-02758-f006:**
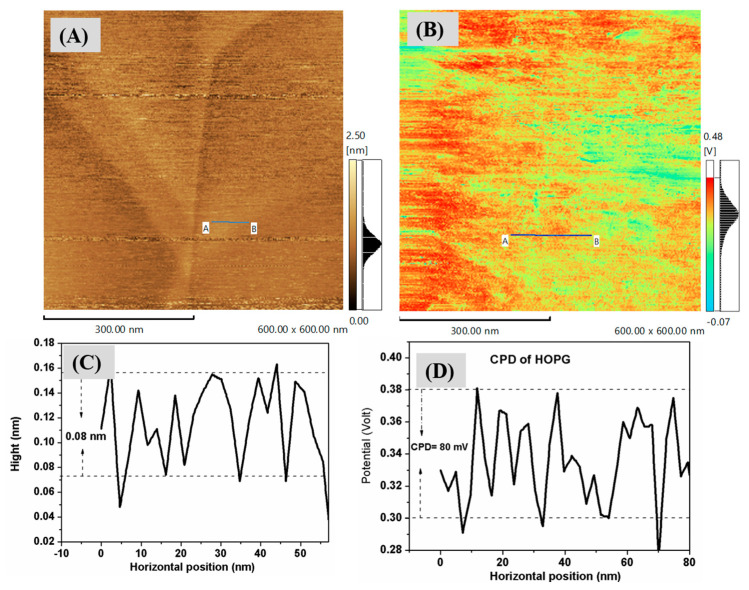
(**A**,**B**) AFM topography and surface potential of reference sample Highly Ordered Pyrolytic Graphite (HOPG), grade ZYA with mosaic spread 0.4 ± 0.1°, chip size 10 mm × 10 mm, and thickness 1 mm (MikroMasch) images, (**C**) height profile of Ag nanoclusters extract from image (**A**) and (**D**) CPD profile measurements extract from image (**B**).

**Figure 7 nanomaterials-13-02758-f007:**
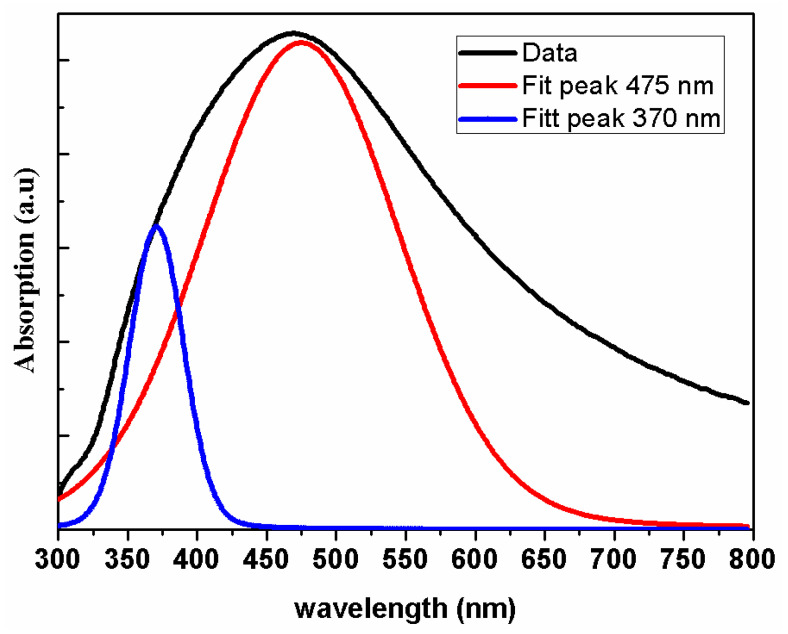
Deconvolution results of room temperature UV-Visible absorption spectra of Ag nanocluster.

**Figure 8 nanomaterials-13-02758-f008:**
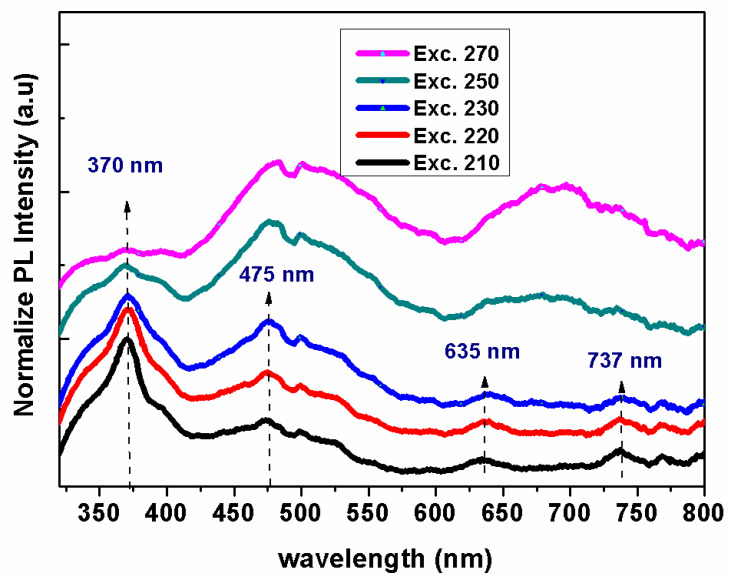
Photoluminescence of silver nanoclusters spectra at different excitation wavelengths (210 nm, 220 nm, 230, 250 nm, and 270 nm).

## Data Availability

The data that support the findings of this study are available from the corresponding authors upon request.

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
