# Peer review of "Ag Nanocluster Production through DC Magnetron Sputtering and Inert Gas Condensation: A Study of Structural, Kelvin Probe Force Microscopy, and Optical Properties"

_nanomaterials, 2023, doi:10.3390/nano13202758_

Round 1
Reviewer 1 Report
The synthesis of nanocrystals up to 10 nm in size is a current trend in modern nanotechnology. The most interesting methods in this case are those using a gas discharge. The authors in the presented work synthesized silver nanoparticles and carried out their thorough analysis.
The work is interesting, however I have some comments:
1) Please provide links to new works devoted to the study of various types of discharges and their application in the synthesis of nanostructures:
a) https://doi.org/10.3390/nano13131966
b) DOI: 10.1088/1361-6595/ac89a7
2) In addition, it is necessary to mention modern works devoted to magnetron sputtering
https://doi.org/10.1021/acs.jpcc.0c09746
and other works
3) In addition, the authors should more clearly define the place of their work among other works performed in this direction. In particular, to more clearly demonstrate novelty.
4) Figures 4 a,b and 5 a,b should be improved. The images are of rather poor resolution and have small signatures.
Reviewer 2 Report
The work present Ag nano cluster synthesized by DC magneto sputtering. The Ag nano clusters were characterized by XRD, SPM, KPFM, etc.
1. The objectives of this work is very vague. A brief search revealed that the synthesis of Ag nanoparticles via DC magnetron sputtering has been extensively researched. For example, Growth of Silver Nanoparticles by DC Magnetron Sputtering' by P. Asanithi, S. Chaiyakun, and P. Limsuwan, published in the Journal of Nano Materials (https://doi.org/10.1155/2012/963609), 'Surface-enhanced Raman scattering substrate of silver nanoparticles depositing on AAO template fabricated by magnetron sputtering' by N. Nuntawong, M. Horprathum, P. Eiamchai, K. Wong-ek, and V. Patthanasettakul published in Vacuum in 2010 (https://doi.org/10.1016/j.vacuum.2009.12.020). It remains unclear from the introduction why this study is better to these publications, which are more than a decade old.
2.Materials and Methods are clearly documented with details.
3.In Section 3.1, if the authors explored the optimization of synthesis conditions, such as different substrates mentioned in line 69, pressures mentioned in line 73, Ar gas in line 89, or discharge power in line 93, a comparison between Ag nanoclusters obtained under different conditions should be provided. However, the authors only present Ag nano clusters of approximately 4 nm in diameter under one set of conditions: a discharge power of 50 W, an Argon flow rate of 60 sccm, a chamber pressure of 1.17 × 10−3 mbar, a U/V ratio of 0.09, and an aggregation length of 40 mm. There is no explanation for the selection of this specific condition for further study.
4.In line 139, it is mentioned, "Height analysis, as shown in Figure 2c, reveals that the observed heights of these nanoclusters range from 4 to 6 nm." However, there is no Figure 2c; if this is referring to Figure 1c, then the presented heights are approximately 3.5-4.5 nm.
5.In Section 3.2, if the authors studied the same sample, which exhibits size variations from 2-6 nm in Figure 1, it is essential to explore how these differences in size affect the measured work function. As indicated in line 203, 'The work function value for Ag nanoclusters is higher than the 4.3 eV work function of bulk silver, as reported in existing literature [21].' It is expected that the size variation would influence the work function. It is important to clarify the motivation for measuring the work function of this sample.
6.In line 166, it is stated, 'Figure 3 displays the AFM topography and KPFM images for silver nanocluster…,' which should be corrected to 'Figure 4.'
7.The optical properties discussed in the paper are well-documented in the literature. This is evidenced by the references cited in Section 3.2. The paper should provide a compelling argument for the significance of the presented results in light of this existing literature.
The figure citation in main text should be very carefully revised to actually figure presented in the paper.
Reviewer 3 Report
I read the text of the article interest. I think that the issue taken up is interesting. The weakest point of the article is that the authors only examined one sample of nanoparticles deposited on the substrate. I would be very interested to read an article presenting the issue in a broader range of technological parameters. I also think the paper was prepared carelessly and needs several corrections before its publication. I provide the comments in the form of the following list.
1. Abstract
The summary should be prepared so that it is a stand-alone text. In this regard, I would expand the abbreviation HOPG as other abbreviations are expanded.
2. Introduction, lines 57 - 58
There is no need to list a detailed list of research methods in the Introduction. It is entirely sufficient to list groups of methods.
3. Introduction
In the Introduction section, the authors correctly and in detail describe the issues of nanoparticle fabrication by magnetron sputtering techniques. In my opinion, they should include in this part a brief description of the technique supporting the agglomeration of particles: Inert Gas Condensation. The Introduction completely neglects to mention it.
4. Materials and Methods, line 70, “(…) highly compatible ultra-high system (…)”
I have an impression that something is missing in this expression.
5. Materials and Methods, apparatus diagram
I recommend that the authors consider whether including the apparatus diagram in the current article would be worthwhile. This makes it very easy for readers to perceive it.
6. Materials and Methods, line 93 – 96 “This power level played a crucial role in determining the size and characteristics of the resulting nanoclusters due to modulate this power level to exert control over the growth and nucleation processes [14].”
And what did the authors gain by choosing the 50W value? In addition, the authors should give all the technological parameters for depositing condensate particles on the substrate. The set of parameters should be accurate enough for the experiment to be repeated by any research group.
7. Results and discussion
In my opinion, much information in this section should have been included in the Materials and Methods section, such as AFM tip, pressures, gas flows, KPFM description, etc.
8. Fig.1
The D section of the figure does not have any caption
9. Fig 3
XRD results are not mentioned in the text. Please check carefully the order of the figures also. There are mistakes
10. Line 204: “The work function value for Ag nanoclusters is higher than the 4.3 eV work function of bulk silver, as reported in existing literature [21].”
And what is the value of existing literature? Why do we have to look in another paper for the answer?
11. Caption if the Fig. 4
There are no Q – P points in the image, as the caption suggests
12. Line 267 – 268 “section is not mandatory but can be added to the manuscript if the discussion is unusually long or complex.”
Round 2
Reviewer 3 Report
Dear authors
I thank the authors for responding to all my comments and for the efforts made to improve the article. The article has no factual errors, which entitles it to be published. Nevertheless, I think it still needs editorial efforts to meet the standards of a scientific text. Here is a list of my comments
1. The Introduction section aims for the authors to justify their choice of methods to achieve their research goals. This includes a write-up of the rationale for using an additional technique to support magnetron sputtering. Mainly when the authors write explicitly, “The advantages of using DC magnetron sputtering for Ag nanocluster synthesis include its ability to produce nanoparticles with controlled size, shape, and density.” If this is the case, what is the additional technique for? The introduction should give the background of the issue in detail. In this case, there are obvious gaps.
In the Materials and Methods section, only the apparatus’s technical data and the parameters for implementing technological processes should be given. This section should not serve as a presentation of the rationale for choosing methods. Writing texts have a specific arrangement of sections for more straightforward navigation of readers and should be adhered to. Readers first!
1. Materials and Methods, line 70, “(…) highly compatible ultra-high system (…)”
I guess the authors meant the “(…) highly compatible ultra-high system vacuum system (…)”
2. Materials and Methods, lack of the apparatus diagram
The authors said: We have shown the apparatus diagram in our previous work that has been cited in this manuscript [Ref. 13 & 16]. In fact, the main purpose of this study is not to show the Ag nanoparticles’ fabrication technique rather studying its properties. Therefore, we see no need to show it again as long as our previous work is cited and interested readers can visit those articles.
Since Ref. 13 and 16 do not describe the technology for producing Ag nanoparticles, they cannot serve to prop up the authors to omit technological details of the process used in the current article. Although the authors focus on the study of properties in this article, the material for the study must be described in detail. If the authors did not purchase the nanoparticles on the market and provide product details, or if they did not produce them in an adequately described earlier experiment, it must be done in this article. This article has the characteristics of a stand-alone paper and should include relevant information.
3. Materials and Methods, line 93 – 96 “This power level played a crucial role in determining the size and characteristics of the resulting nanoclusters due to modulate this power level to exert control over the growth and nucleation processes [14].”
My question was, what is behind the fact that 50W power is favorable in nanoparticle synthesis? I asked this question because the power value is not an arbitrary parameter. Moreover, since it appears in the text, I would prefer to learn more about why this condition is essential. By the way, it is better to use power density when characterizing magnetron sputtering.
4. R: In my opinion, much information in this section should have been included in the Materials and Methods section, such as AFM tip, pressures, gas flows, KPFM description, etc.
A: The opinion of the reviewer is well respected, but we believe is good to keep it the
results section, and many authors have done the similar way.
The argument that something is right because many people are doing something is not convincing. As I wrote earlier, the division and arrangement of sections in written texts has a purpose. Looking for random information in random places in the text is exceptionally annoying (not to mention other articles)
